# RAPID: An Efficient Reinforcement Learning Algorithm for Small Language Models

## Abstract

Reinforcement learning (RL) has emerged as a promising strategy for finetuning small language models (SLMs) to solve targeted tasks such as math and coding. However, RL algorithms tend to be resource-intensive, taking a significant amount of time to train. We propose RAPID, a novel RL algorithm that can substantially reduce the running time of RL. Our key insight is that RL tends to be costly due to the need to perform both inference and backpropagation during training. To maximize use of computational resources, our algorithm performs inference in large batches, and then performs off-policy policy gradient updates in mini-batches. For off-policy updates, we incorporate group advantage estimation into the policy gradient algorithm, and derive an importance weighted estimator to correct for the bias arising from off-policy learning. Our experiments demonstrate that our algorithm can reduce running time by 11%–34% on three benchmarks compared to state-of-the-art RL algorithms while maintaining similar or better accuracy.

## 1 Introduction

Small language models (SLMs) are increasingly being adopted in resource-constrained environments such as phones, laptops, and mobile robots. In many applications, SLMs can be finetuned to improve their capabilities at a specific task. While distillation from larger models is a common strategy, it can have limited effectiveness in many domains since the labeled outputs deviate far from the distribution of outputs produced by the SLM. Reinforcement learning (RL) provides a promising alternative, training the SLM based on its own successful and unsuccessful generations.

However, RL algorithms tend to be significantly more compute intensive compared to distillation. The key challenge is the need to orchestrate inference and backpropagation. One option is to allocate one subset of GPUs to inference and another to backpropagation, balancing them to maximize utilization of each GPU (Hu et al., 2025; Espeholt et al., 2018). However, this approach work best when there are a large number of GPUs, enabling the system to keep all GPUs busy. A simpler approach is to alternate between inference and training, using all GPUs for each phase. However, this alternation can introduce substantial latency switching between inference and backpropagation.

We propose to scale this approach by substantially increasing the size of each batch sampled during an inference phase, and then taking multiple gradient steps on this batch using off-policy RL. The key question is the choice of off-policy RL algorithm. A standard choice would be to use PPO (Schulman et al., 2017b), which relies on KL-divergence regularization; however, we find that this approach limits performance when taking a large number of off-policy gradient steps.

Instead, we build on the original policy gradient algorithm (Sutton et al., 1999), where importance weighting can be used for off-policy corrections. Existing approaches typically train a value function that incorporates off-policy corrections (Espeholt et al., 2018); however, recent work suggests that Monte Carlo estimates of the advantage function at the group level (i.e., for a single math or coding problem) tend to be more effective in for training language models (Shao et al., 2024). We devise an importance weighted version of the policy gradient algorithm under group advantage estimation, and then uses this estimator to make off-policy gradient updates.

We perform an extensive empirical evaluation on three datasets: MBPP+ (Austin et al., 2021; Liu et al., 2023), MATH (Hendrycks et al., 2021; Lightman et al., 2023), and MiniF2F (Zheng et al.,

2022). We show that RAPID reduces training time by 11%–34% without sacrificing accuracy. We also analyze the impact of off-policy sampling on sample staleness, runtime, and accuracy.

To summarize, our key contributions are as follows:

- We propose RAPID (**R**eweighted **A**dvantage for **P**reemptive **I**nference of **D**ata), an RL algorithm that alternates between performing inference in large batches and performing off-policy policy gradient updates in small mini-batches, allowing for separately optimized inference and backpropagation batch sizes to substantially improving runtime.

- We incorporate group advantage estimation into the policy gradient algorithm, and derive an importance weighted estimator to correct for the bias arising from off-policy learning.

- We perform an extensive empirical evaluation, demonstrating that RAPID reduces training time by 11%–34% on three benchmarks generally without sacrificing accuracy.

## 2 RELATED WORK

**Improving LM Reasoning Using RL.** Given the promising performance of language models (LMs), numerous studies have explored their application to mathematical problem solving (Hendrycks et al., 2021; Cobbe et al., 2021; Glazer et al., 2024), program synthesis (Austin et al., 2021; Puri et al., 2021), and other reasoning tasks. For instance, Chain-of-Thought prompting (Wei et al., 2023) encourages LMs to generate intermediate reasoning steps before producing the final answer. Tree-of-Thought (Yao et al., 2023) and Graph-of-Thought (Besta et al., 2024) extend this idea by imposing logical structure to organize the reasoning process. Recent efforts have focused on using RL to improve LM reasoning capabilities. In question-answering tasks, FireAct (Chen et al., 2023) and AgentTuning (Zeng et al., 2023) enhance reasoning capabilities by learning from demonstrations from humans or stronger models. These approaches are commonly referred to as *supervised fine-tuning* (SFT), or *behavior cloning* in the RL literature. However, several studies have found limits on the effectiveness of SFT, and significantly improved LLM reasoning capability via on-policy RL (DeepSeek-AI et al., 2025a; Shao et al., 2024; Zeng et al., 2025).

**Efficient RL.** Although on-policy RL algorithms can effectively improve LM reasoning capability, they can be very computationally expensive, since they require re-sampling generations after each model update. To mitigate this issue, DeepSeek-AI et al. (2025b) propose more efficient transformer architectures to accelerate pretraining, and Kwon et al. (2023b) introduce advanced memory management techniques to speed up sampling in post-training. Although current RL algorithms can leverage vLLM acceleration, the full potential of vLLM remains underutilized, leaving significant room for improving RL efficiency. Furthermore, efficient RL frameworks have also been proposed. For instance, VeRL (Sheng et al., 2025) utilizes sing-controller to coordinate inter-node data resharding, and multi-controller within intra-node computation; and OpenRLHF (Hu et al., 2025) proposes an open-source and scalable RLHF framework. We note that these works improve RL efficiency by facilitating better sharding and parallel computation, which is complementary to our algorithmic contributions; furthermore, they largely target settings with a large amount of computational resources, whereas we focus on more resource constrained settings.

**Off-Policy and Offline RL.** The efficiency of RL can also be improved via offline and off-policy RL algorithms. Offline algorithms such as SFT and Direct Preference Optimization (DPO) (Rafailov et al., 2024) rely entirely on offline data (i.e., no inference during training), meaning standard supervised learning frameworks can be used for efficient training. However, offline algorithms usually obtain suboptimal performance due to the lack of online exploration. Off-Policy RL algorithms sample new data during training, but possibly using a stale policy (i.e., different from the policy whose gradient is being taken). For instance, PPO (Schulman et al., 2017b), TRPO (Schulman et al., 2017a), and GRPO (Shao et al., 2024) take multiple gradient steps on data sampled from a stale policy, using KL-divergence constraints or regularization to mitigate off-policy bias. Another strategy is to use importance weighting to correct the standard policy gradient estimator (Sutton et al., 1999). Empirically, we find that this strategy works better than using KL-divergence regularization. Our off-policy RL algorithm builds on the latter, combining importance weighting with group advantage estimation that is standard in RL for language modeling tasks (Shao et al., 2024).

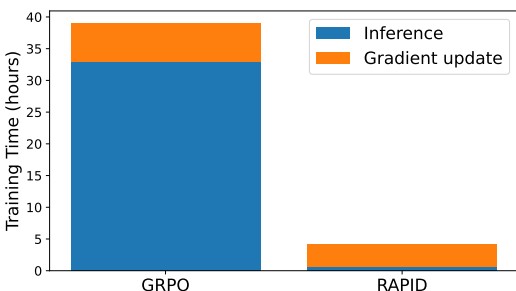

Figure 1: RAPID can reduce running time by 89% compared to a naïve GRPO that uses the same inference and training batch size with separate GPUs for inference and backpropagation.

## 3 RAPID ALGORITHM

### 3.1 PRELIMINARIES

We consider an SLM $\pi_\theta$ with parameters $\theta$, which takes in a user prompt $\mathbf{x}$ and generates output $\mathbf{y}$; we let $y_t$ denote the $t$th token in $\mathbf{y}$. We assume given a prompt training set $X = \{\mathbf{x}_n\}_{n=1}^N$ (e.g., math or coding problems). For a training prompt $\mathbf{x}_n$, we can check whether an output $\mathbf{y}$ produces the correct answer, represented as a scalar reward $R(\mathbf{x}_n, \mathbf{y}_n) \in \mathbb{R}$. We assume that this reward is given once the output is fully generated (e.g., a binary indicator of whether the final answer is correct); while this precludes process rewards (Wang et al., 2024), recent work has found that they may not be effective in our setting due to the difficulty predicting whether an output is on the right track (DeepSeek-AI et al., 2025a). For simplicity, we assume $R$ is deterministic (which is typical for tasks such as math and coding), but our approach extends without modification to stochastic rewards. Now, our goal is to compute the policy $\pi_\theta(\mathbf{y}_n \mid \mathbf{x}_n)$ that maximizes expected reward:

$$\pi_{\theta^*} = \arg\max_\theta J(\theta) \qquad \text{where} \qquad J(\theta) = \frac{1}{N} \sum_{x_n \in X} \mathbb{E}_{\mathbf{y}_n \sim \pi_\theta(\cdot | \mathbf{x}_n)}[R(\mathbf{x}_n, \mathbf{y}_n)].$$

This RL problem can be viewed as a one-step MDP (i.e., a contextual bandit), where the initial state is a prompt $\mathbf{x}_n$, the action is the generation $\mathbf{y}_n$, and the reward is $R(\mathbf{x}_n, \mathbf{y}_n)$.

### 3.2 ALTERNATING INFERENCE AND BACKPROPAGATION

A key feature of RL for LMs is that inference typically occurs on specialized inference servers such as vLLM (Kwon et al., 2023a). Importantly, inference is typically much more memory efficient than backpropagation, meaning much larger batches are optimal for inference compared to backpropagation. Empirically, sampling takes up a much larger portion of training time than backpropagation if performed in small batches. Figure 1 compares our algorithm to running GRPO with separate inference and backpropagation GPUs with the same batch size, and shows time taken for inference vs. backpropagation; as can be seen, naïvely running GRPO this way is very slow. Instead, our algorithm alternates between inference and backpropagation, using all GPUs for each phase.

Our algorithm assumes given a set of training prompts $X$. During the $t$th outer step, it samples a large number $N_{\text{inference}}/N_{\text{group}}$ of prompts $X_t$ from $X$, samples $N_{\text{group}}$ outputs $\mathbf{y}_{n,i} \sim \pi_\theta(\cdot \mid \mathbf{x}_n)$ for each $\mathbf{x}_n \in X_t$, and collects these examples to form a training dataset $Z_t$ with $N_{\text{inference}}$ examples. Given $Z_t$, it then performs $H = N_{\text{inference}}/N_{\text{step}}$ policy-gradient updates to $\pi_\theta$ with an off-policy RL algorithm, each update using rewards from a mini-batch of $N_{\text{step}}$ samples drawn from $Z_t$. By taking a large number of samples during inference, this algorithm maximizes efficiency of the inference server; by performing backpropagation and policy gradient updates on smaller batches, it maximizes the efficiency of gradient descent. Our implementation iterates over prompts $X_t \subseteq X$ in the outer loop and over mini-batches $Z_h \subseteq Z$ in the inner loop rather than using sampling.

The remaining question is what to use as a policy gradient update; we describe the on-policy policy gradient with group advantage estimation in Section 3.3, and how we use importance weighting to adapt it to off-policy learning in Section 3.4. Our full algorithm is summarized in Algorithm 1.

---

**Algorithm 1** RAPID Algorithm

**procedure** RAPID-RL($N_{\text{inference}}, N_{\text{group}}, N_{\text{step}}, X, \pi_\theta$)
    **for** $t \in [T]$ **do**
        Sample an inference batch $X_t$ of size $N_{\text{inference}}/N_{\text{group}}$ from $X$
        Sample $\mathbf{y}_{n,i} \sim \pi_\theta(\cdot \mid \mathbf{x}_n)$ for each $\mathbf{x}_n \in X_t$ and each $i \in [N_{\text{group}}]$
        Form the dataset $Z_t = \{(\mathbf{x}_n, \mathbf{y}_{n,i}) \mid \mathbf{x}_n \in X_t, i \in [N_{\text{group}}]\}$
        Copy the current policy $\mu \leftarrow \pi_\theta$
        Run IW-GRPG($N_{\text{step}}, Z_t, \pi_\theta, \mu$)
    **end for**
**end procedure**
**procedure** IW-GRPG($N, Z, \pi_\theta, \mu$)
    **for** $h \in [H]$ **do**
        Sample a mini-batch $Z_h$ of size $N$ from $Z$
        Compute the advantage estimate $A^{\pi_\theta}(\mathbf{x}_n, \mathbf{y}_n)$ for each $(\mathbf{x}_n, \mathbf{y}_n) \in Z_h$ using Eq. (5)
        Compute the policy gradient $\nabla_\theta J(\theta)$ using $Z_h$ according to Eq. (3)
        Take a gradient ascent step on $\pi_\theta$
    **end for**
**end procedure**

---

### 3.3 GROUP RELATIVE POLICY GRADIENT

Our off-policy RL builds on the policy gradient (PG) algorithm (Sutton et al., 1999). This algorithm is on-policy; we will use importance weighting to convert it into an off-policy algorithm. The Policy Gradient Theorem gives the following way to compute the gradient:

$$\nabla_\theta J(\theta) = \frac{1}{N} \sum_{n=1}^{N} \mathbb{E}_{\mathbf{y}_n \sim \pi_\theta(\cdot|\mathbf{x}_n)} \left[ \frac{\nabla_\theta \pi_\theta(\mathbf{y}_n \mid \mathbf{x}_n)}{\pi_\theta(\mathbf{y}_n \mid \mathbf{x}_n)} A^{\pi_\theta}(\mathbf{x}_n, \mathbf{y}_n) \right], \tag{1}$$

where $A^{\pi_\theta}(\mathbf{x}_n, \mathbf{y}_n)$ is the advantage function (Sutton & Barto, 2018). Note that this update is on-policy since the trajectories $\mathbf{y}_n$ must be sampled using the current policy $\pi_\theta$.

A key question is how to estimate the advantage function $A^\pi(\mathbf{x}_n, \mathbf{y}_n) = Q^\pi(\mathbf{x}_n, \mathbf{y}_n) - V^\pi(\mathbf{x}_n)$, where $Q^\pi$ is the Q-function and $V^\pi$ is the value function. In our setting, recent work has found that training value and $Q$-function estimators is highly biased due to the difficulty in predicting $V^\pi$ and $Q^\pi$ for language modeling tasks (Liang et al., 2022). Thus, we focus on Monte Carlo approaches. The most popular Monte Carlo approach is the *single-path method*, which uses the estimate $A^{\pi_\theta}(\mathbf{x}_n, \mathbf{y}_n) \approx R(\mathbf{x}_n, \mathbf{y}_n) - b$, where $b = N^{-1} \sum_{n'=1}^{N} R(\mathbf{x}_{n'}, \mathbf{y}_{n'})$ is the baseline. A shortcoming is that $b$ is an estimate of the average value across all examples $n' \in [N]$, so $b$ estimates the average value function $N^{-1} \sum_{n=1}^{N} V(\mathbf{x}_n)$ rather than the value $V(\mathbf{x}_n)$ for the current prompt. The *vine method* (Kazemnejad et al., 2024; Schulman et al., 2017a) uses targeted sampling to fix this issue; however, it requires a large number of samples, making it computationally infeasible.

Instead, recent work has propose *group* advantage estimation, which interpolates between the single-path and vine methods. It exploits the fact that in the reasoning setting, we typically train on multiple samples $\mathbf{y}_n$ for a single user prompt $\mathbf{x}_n$. In our formulation, we can think of there being multiple $\mathbf{x}_n$ that are identical. Suppose that we partition $[N] = \{1, ..., N\}$ into groups $\mathcal{N}_1, ..., \mathcal{N}_K$, where $\mathbf{x}_n$ is the same for all $n \in \mathcal{N}_k$. Then, it estimates the advantage using the formula

$$A^{\pi_\theta}(\mathbf{x}_n, \mathbf{y}_n) \approx R(\mathbf{x}_n, \mathbf{y}_n) - \frac{1}{N_k} \sum_{n' \in \mathcal{N}_k} R(\mathbf{x}_n, \mathbf{y}_{n'}), \tag{2}$$

where $\mathcal{N}_k$ is the group containing $n$ and $N_k = |\mathcal{N}_k|$. This estimate is based on two facts. First, in our setting, the $Q$-function is simply the reward $Q^\pi(\mathbf{x}_n, \mathbf{y}_n) = R(\mathbf{x}_n, \mathbf{y}_n)$. Second, it uses samples in the group $\mathcal{N}_k$ to form an estimate of the value function at $\mathbf{x}_n$ (since $\mathbf{x}_{n'} = \mathbf{x}_n$ for all $n' \in \mathcal{N}_k$):

$$V^{\pi_\theta}(\mathbf{x}_n) = \mathbb{E}_{\mathbf{y}_{n'} \sim \pi_\theta(\cdot|\mathbf{x}_n)}[R(\mathbf{x}_n, \mathbf{y}_{n'})] \approx \frac{1}{N_k} \sum_{n' \in \mathcal{N}_k} R(\mathbf{x}_n, \mathbf{y}_{n'}).$$

This estimate can also be viewed as performing a vine estimate of the advantage at $\mathbf{x}_n$. We refer to the algorithm combining group advantage estimation and the policy gradient algorithm as group relative policy gradient (GRPG); this algorithm is a traditional on-policy RL algorithm.

### 3.4 Importance Weighted Group Relative Policy Gradient

We propose an off-policy modification of GRPG. Specifically, suppose we have a behavioral policy $\mu(\cdot \mid \mathbf{x}_n)$ from which we are collecting data. First, we can form the standard importance weighted estimate of the policy gradient, assuming for now that we know the advantage function:

$$\nabla_\theta J(\theta) = \frac{1}{N} \sum_{n=1}^{N} \mathbb{E}_{\mathbf{y}_n \sim \mu(\cdot \mid \mathbf{x}_n)} \left[ \frac{\nabla_\theta \pi_\theta(\mathbf{y}_n \mid \mathbf{x}_n)}{\mu(\mathbf{y}_n \mid \mathbf{x}_n)} A^{\pi_\theta}(\mathbf{x}_n, \mathbf{y}_n) \right]. \tag{3}$$

However, we need some way to estimate the advantage function from off-policy samples. To this end, we form an importance weighted version of the group advantage estimate in Eq. (2). First, the $Q$-function does not depend on $\pi_\theta$, so we have $Q^{\pi_\theta}(\mathbf{y}_n \mid \mathbf{x}_n) = R(\mathbf{x}_n, \mathbf{y}_n)$. Second, the value function estimate is importance weighted to account for the fact that the samples now come from $\mu$:

$$V^{\pi_\theta}(\mathbf{x}_n) = \mathbb{E}_{\mathbf{y}_{n'} \sim \mu(\cdot \mid \mathbf{x}_n)} \left[ \frac{\pi_\theta(\mathbf{y}_n \mid \mathbf{x}_n)}{\mu(\mathbf{y}_n \mid \mathbf{x}_n)} R(\mathbf{x}_n, \mathbf{y}_{n'}) \right]$$

$$\approx \frac{1}{N_k} \sum_{n' \in \mathcal{N}_k} \frac{\pi_\theta(\mathbf{y}_{n'} \mid \mathbf{x}_n)}{\mu(\mathbf{y}_{n'} \mid \mathbf{x}_n)} R(\mathbf{x}_n, \mathbf{y}_{n'}).$$

Putting these two estimates together, the importance weighted group advantage estimate is

$$A^{\pi_\theta}(\mathbf{x}_n, \mathbf{y}_n) \approx R(\mathbf{x}_n, \mathbf{y}_n) - \frac{1}{N_k} \sum_{n' \in \mathcal{N}_k}^{N} \frac{\pi_\theta(\mathbf{y}_{n'} \mid \mathbf{x}_{n'})}{\mu(\mathbf{y}_{n'} \mid \mathbf{x}_{n'})} R(\mathbf{x}_{n'}, \mathbf{y}_{n'}). \tag{4}$$

Our importance weighted GRPG algorithm uses the advantage estimate Eq. (4) in the importance weighted policy gradient Eq. (3). The resulting gradient is almost an unbiased estimate of $\nabla_\theta J(\pi_\theta)$, except for the fact that the $(\mathbf{x}_n, \mathbf{y}_n)$ is reused in the $n$th summand of $\nabla_\theta J(\theta)$ and in $V^{\pi_\theta}(\mathbf{x}_n)$; following Ahmadian et al. (2024), we can fix it by summing over $\mathcal{N}_k \setminus \{n\}$ instead of $\mathcal{N}_k$:

$$A^{\pi_\theta}(\mathbf{x}_n, \mathbf{y}_n) \approx R(\mathbf{x}_n, \mathbf{y}_n) - \frac{1}{N_k} \sum_{n' \in \mathcal{N}_k \setminus \{n\}}^{N} \frac{\pi_\theta(\mathbf{y}_{n'} \mid \mathbf{x}_{n'})}{\mu(\mathbf{y}_{n'} \mid \mathbf{x}_{n'})} R(\mathbf{x}_{n'}, \mathbf{y}_{n'}).$$

Using this advantage estimate instead of Eq. (4) results in an unbiased estimate of $\nabla_\theta J(\theta)$; however, since the bias is small when $N_k$ is large, we use Eq. (4) to keep our algorithm simple. Finally, to promote stability, we implement standard importance weight clipping:

$$A^{\pi_\theta}(\mathbf{x}_n, \mathbf{y}_n) \approx R(\mathbf{x}_n, \mathbf{y}_n) - \frac{1}{N_k} \sum_{n' \in \mathcal{N}_k}^{N} \max \left\{ \frac{\pi_\theta(\mathbf{y}_{n'} \mid \mathbf{x}_{n'})}{\mu(\mathbf{y}_{n'} \mid \mathbf{x}_{n'})}, \eta \right\} R(\mathbf{x}_{n'}, \mathbf{y}_{n'}). \tag{5}$$

While this strategy can introduce bias into the gradient estimate, it significantly reduces variance in the gradient estimate due to large importance weights.

## 4 Experiments

We empirically evaluate RAPID compared to several RL baselines, with the following key results:

- We show that RAPID achieves comparable accuracy to baselines while significantly reducing runtime across several datasets, model sizes, and model families (Section 4.2).

- We analyze the influence of off-policy sampling on sample staleness, runtime, and accuracy, with a representative qualitative example of token-level staleness for a single generation, as well as an analysis of importance weight clipping (Section 4.3).

### 4.1 Experimental Setup

**Baselines.** We compare our approach to SFT (Ouyang et al., 2022), GRPO (Shao et al., 2024), Policy Gradient (Sutton et al., 1999), and DAPO (Yu et al., 2025).

| Algorithm | MBPP+ | | | MATH | | MiniF2F | |
|---|---|---|---|---|---|---|---|
| | Pass@1 | Pass@8 | Runtime | Pass@1 | Runtime | Pass@1 | Runtime |
| Base | 3.2% | 24.3% | N/A | 22.4% | N/A | 4.0% | N/A |
| | ± 1.0% | ± 2.2% | N/A | ± 0.2% | N/A | ± 0.2% | N/A |
| SFT | 5.6% | 35.4% | 2.0m | 21.0% | 4h42m | 4.5% | 0.3m |
| | ± 2.8% | ± 1.3% | ± 0.0m | ± 0.7% | ± 9m | ± 0.0% | ± 0m |
| GRPO | 6.4% | 37.1% | 22.0m | 23.2% | 5h36m | 5.5% | 51.1m |
| | ± 3.5% | ± 1.8% | ± 1.5m | ± 0.1% | ± 2m | ± 0.5% | ± 10.4m |
| PG | 6.4% | 39.5% | 19.2m | **30.6%** | 4h49m | **6.6%** | 41.1m |
| | ± 2.0% | ± 1.8% | ± 1.6m | ± 0.6% | ± 6m | ± 0.0% | ± 0.5m |
| DAPO | 5.3% | 32.5% | 19.2m | 29.5% | 4h47m | 4.8% | 49.0m |
| | ± 2.3% | ± 0.9% | ± 0.7m | ± 0.5% | ± 9m | ± 1.3% | ± 6.4m |
| RAPID ($H = 8$) | **11.1%** | 38.6% | **12.7m** | 28.7% | 4h29m | 5.6% | **27.9m** |
| | ± 2.0% | ± 0.9% | ± 0.6m | ± 0.8% | ± 9m | ± 0.2% | ± 5.5m |
| RAPID ($H = 4$) | 9.9% | **47.4%** | 13.3m | 29.5% | **4h14m** | 4.8% | 31.1m |
| | ± 1.0% | ± 0.0% | ± 1.1m | ± 0.3% | ± 0m | ± 0.2% | ± 5.6m |

Table 1: Accuracy and runtime of RL algorithms across our datasets on Qwen2.5-0.5B. Best performing results under each metric are in bold, runner-ups are underlined (SFT is not included in the runtime comparison because of its poor accuracy compared to other approaches).

| Algorithm | Qwen 2.5 | | | Llama 3.2 | | | Gemma 3 | | |
|---|---|---|---|---|---|---|---|---|---|
| | 1.5B | | | 1B | | | 1B | | |
| | Pass@1 | Pass@8 | Time | Pass@1 | Pass@8 | Time | Pass@1 | Pass@8 | Time |
| Base | 7.9% | 57.0% | N/A | 3.5% | 25.4% | N/A | 0.9% | 5.3% | N/A |
| SFT | 13.2% | 58.8% | 4.0m | 7.0% | 22.8% | 3.3m | 0.0% | 0.0% | 4.0m |
| GRPO | 19.3% | 64.9% | 33.3m | 7.9% | 33.3% | 27.4m | 0.0% | 10.5% | 48.3m |
| PG | **32.5%** | **66.7%** | 24.7m | 18.4% | 42.1% | 18.5m | 0.9% | 10.5% | 31.6m |
| DAPO | 16.7% | 64.0% | 36.5m | **21.1%** | 40.4% | 25.5m | 1.8% | 12.3% | 40.9m |
| RAPID ($H = 4$) | 28.9% | **66.7%** | **22.8m** | 14.9% | **44.7%** | **15.2m** | **3.5%** | **14.0%** | **24.4m** |

Table 2: Performance of different algorithms across model families and sizes on MBPP+. Best performing results under each metric are in bold, runner-ups are underlined (SFT is not included in the runtime comparison because of its poor accuracy compared to other approaches).

**Models.** We consider small models including Qwen-2.5 0.5B and 1.5B (Qwen et al., 2025), Llama-3.2 1B (Grattafiori et al., 2024), and Gemma-3 1B (Team et al., 2025).

**Datasets.** We perform our experiments on three datasets: MBPP+ for coding (264 training and 114 evaluation) (Austin et al., 2021; Liu et al., 2023), MATH (12,000 training and 500 evaluation) (Hendrycks et al., 2021; Lightman et al., 2023) for mathematics, and MiniF2F (244 training and 244 evaluation) (Zheng et al., 2022) for formal theorem-proving.

**Metrics.** For MBPP+, we report both Pass@1 and Pass@8 (which is standard for this dataset); for MATH and MiniF2F, we simply report Pass@1 (i.e., accuracy).

**Compute.** All runs are performed on $4\times$ Nvidia RTX A6000 GPUs on a single server.

**Hyperparameters.** Our main hyperparameters for RAPID are the inference batch size $N_{\text{inference}}$ and the training batch size $N_{\text{step}}$. We report results for different values of the *batch size ratio* $H = N_{\text{inference}}/N_{\text{step}}$, with the training batch size $N_{\text{step}}$ fixed for each dataset (see Appendix A). Appendix A includes additional details on our choices of hyperparameters and configurations.

| $H$ | MBPP+ | | | | MATH | | | MiniF2F | | |
|---|---|---|---|---|---|---|---|---|---|---|
| | Pass@1 | Pass@8 | Time | Stale | Pass@1 | Time | Stale | Pass@1 | Time | Stale |
| 2 | 10.2% ± 2.8% | 41.5% ± 2.2% | 15.5m ± 0.4m | 0.06 ± 0.01 | 29.4% ± 0.6% | 4h47m ± 1.7m | 0.03 ± 0.00 | 5.1% ± 0.5% | 36.4m ± 5.6m | 0.11 ± 0.03 |
| 4 | 9.9% ± 1.0% | **47.4%** ± 0.0% | 13.3m ± 1.1m | 0.08 ± 0.02 | **29.5%** ± 0.3% | 4h14m ± 0.0m | 0.04 ± 0.00 | 4.8% ± 0.2% | 31.1m ± 5.6m | 0.15 ± 0.03 |
| 8 | 11.1% ± 2.0% | 38.6% ± 0.9% | **12.7m** ± 0.6m | 0.11 ± 0.02 | 28.7% ± 0.8% | 4h29m ± 8.7m | 0.06 ± 0.00 | **5.6%** ± 0.2% | 27.9m ± 5.5m | 0.22 ± 0.01 |
| 16 | **11.4%** ± 1.5% | 38.9% ± 1.3% | 13.6m ± 1.0m | 0.13 ± 0.01 | 28.3% ± 0.8% | **4h8m** ± 8.1m | 0.07 ± 0.01 | 4.0% ± 1.3% | **26.2m** ± 5.5m | 0.25 ± 0.08 |

Table 3: Staleness analysis across datasets for Qwen2.5-0.5B. Best performing results under each metric are in bold, runner-ups are underlined. Recommended choices of $H$ are boxed.

| Dataset | Metric | $r(k, \text{Acc})$ | $r(k, \text{Time})$ | $r(k, \text{Stale})$ |
|---|---|---|---|---|
| MBPP+ | Pass@1 | 0.311 | −0.388 | 0.885 |
| | Pass@8 | −0.559 | −0.388 | 0.885 |
| MATH | Pass@1 | −0.648 | −0.666 | 0.936 |
| MiniF2F | Pass@1 | −0.534 | −0.594 | 0.758 |

Table 4: Pearson correlations between batch size ratio $H$ and {accuracy, runtime, and staleness}.

## 4.2 PERFORMANCE OF RAPID

We compare RAPID with $H \in \{4, 8\}$ to our baselines on each of our datasets. We perform three runs for each setup in Table 1 with different random seeds and show means and standard deviations. We find that RAPID reduces training time by **34% for MBPP+, 32% for MiniF2F, and 11% for MATH** when compared to the strongest baseline, while maintaining similar or better accuracy (Table 1). In several cases, RAPID outperforms all baselines in accuracy, while in others RAPID is close to the optimal especially given the small sizes of the datasets. Our results generalize well to different model sizes and families (Table 2). We note that RAPID performs particularly well for pass@8 with MBPP+, outperforming all baselines in time *and* accuracy for all models sizes and families tested here. We hypothesize this is due to the fact that off-policy training can lead to higher diversity in generations, either by increasing entropy in their sampling distributions or by exposing the model to different modes of generations, such as that of previous policies (Shypula et al., 2025).

## 4.3 STALENESS AND IMPORTANCE WEIGHT ANALYSIS

To quantify the effect of off-policy sampling on training accuracy and time, we run RAPID with batch size ratio $H \in \{2, 4, 8, 16\}$. We measure sample staleness in each case by averaging the clipped importance weight in log scale across all steps (Table 3). Again, we perform three runs for each setup and report mean and standard deviation. We then calculate the Pearson correlations between batch size ratio and {staleness, runtime, and accuracy} (Table 4).

**Relating $H$ and staleness.** There is strong positive correlation between batch size ratio and staleness, which is expected since batch size ratio reflects how much *ahead* the inference policy samples in the dataset, which in turn leads to staleness of these samples in subsequent gradient updates.

**Relating $H$ and runtime.** There is negative correlation between batch size ratio and runtime. This effect is because larger inference batch sizes provide amortized inference time per sample, and the much larger optimal batch size for inference compared to backpropagation should justify large batch sampling (Section 3.2). Scenarios where runtime does not decrease as a function of batch size ratio $H$ (e.g., in MATH from $H = 4$ to $H = 8$) are due to longer generations with $H = 8$, which lead to longer inference time (Figure 2). This effect may be because $H = 8$ causes more off-policy training than $H = 4$, leading to verbose generations due to distribution shift from inference to the current

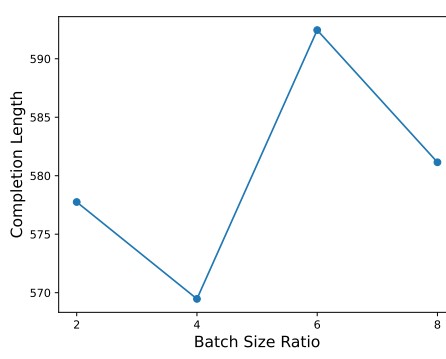 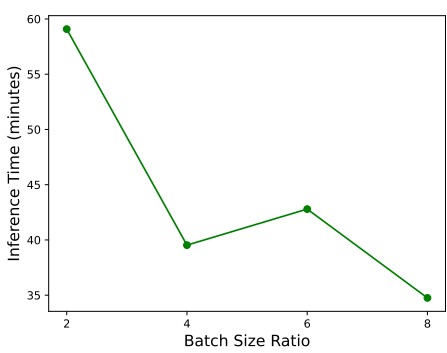

(a) Completion length v.s. batch size ratio.    (b) Inference time v.s. batch size ratio.

Figure 2: We show how completion length and inference time vary with batch size ratio $H$ for Qwen2.5-0.5B on MATH; $H = 8$ leads to longer generation and inference time than $H = 4$.

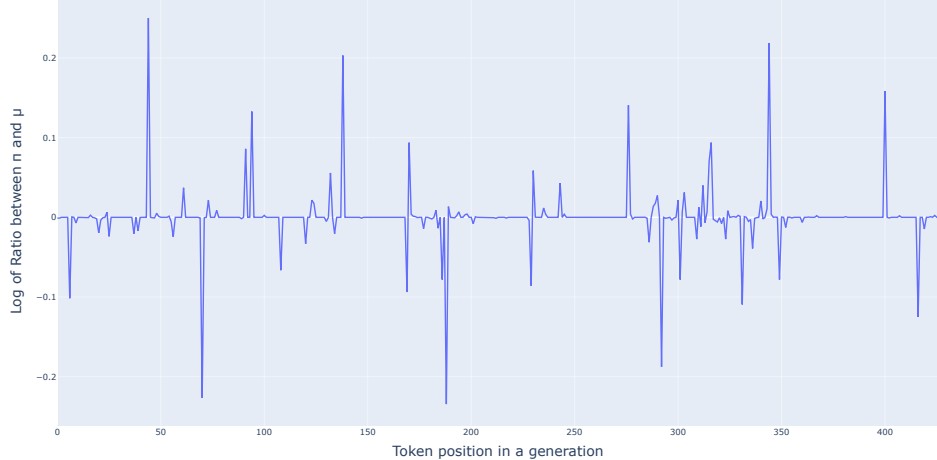

Figure 3: We show the token-level importance weight in log scale for a single generation during training. We observe that other samples encountered throughout training exhibit similar patterns.

policy. Note that the generation is longer still for $H = 16$ compared to $H = 4$ for MATH, but the amortized time benefit for large batch inference makes up for the increase in length. Overall, these results call for a careful tuning of $H$ as increasing it does not necessarily reduce training time. We highlight the optimal training times in Table 3.

**Relating $H$ and accuracy.** The effect of batch size ratio on accuracy is varied across datasets: in most cases there is negative correlation, since larger batch ratios bring samples further off-policy, which can indeed affect final accuracy. Surprisingly, for Pass@1 of MBPP+, there is some positive correlation between accuracy and batch size ratio, leading to both improved accuracy and shorter training time for the larger ratios.

**Optimal choice of $H$.** Overall, batch size ratio $H$ is a tunable parameter that has varied trade-offs on training accuracy and time for different datasets, accuracy metrics, model sizes, and families. We highlight the best $H$ for each dataset and metric in Table 3.

**Token-level importance weight.** Qualitatively, difference in sample probabilities between $\mu$ and $\pi$ almost always occurs on the level of individual tokens or short combinations of tokens. Figure 3 shows a typical example of the token-level importance weight in log scale from a sample generation. This pattern is general across samples and throughout training.

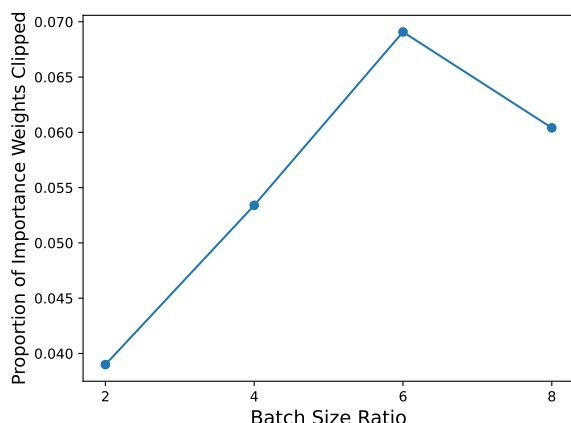

Figure 4: Proportion of importance weights that are clipped for different batch size ratios on Qwen2.5-0.5B for MATH.

**Importance-weight clipping.** Finally, we report the proportion of importance weights clipped for Qwen2.5-0.5B on MATH. Importance weight clipping is applied on the generation level after accumulating per-token importance weights from each generation (Eq. (5)). They average to around 5% as shown in Figure 4. We emphasize that clipping is crucial to prevent cumulative blow-up of token-level importance weights at the generation level, as is evident in our training.

## 5  CONCLUSION

We have proposed RAPID, an RL algorithm that alternates between performing inference in large batches and performing off-policy gradient updates in smaller mini-batches, allowing for a optimal batch sizes for inference and backpropagation. Furthermore, we derived an importance weighted policy gradient estimator to correct for the bias arising from off-policy learning. Our experiments demonstrated that RAPID reduces training time by 11%–34% on three benchmarks while maintaining similar or better accuracy. Finally, we analyzed the influence of off-policy sampling on sample staleness, accuracy, and training time. We provided a representative qualitative example of the token-level importance weights of a single generation, and demonstrated the effects of generation-level importance weight clipping. While our study focuses on SLMs in resource constrained setting, our algorithm may have benefits for training larger models as well.

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

## A   EXPERIMENTAL DETAILS

Our implementation is based on Huggingface's TRL library (von Werra et al., 2020). We list the hyperparameters used in Table 5 for SFT, in Table 6 for RL algorithms. We also include our package versions (Table 7) and DeepSpeed configuration (Rajbhandari et al., 2020) (Figure 5) for full reproducibility.

| Teacher model hyperparameter | |
| --- | --- |
| Teacher Model | Qwen2.5-32B-Instruct |
| Max generation length | 2048 |
| Temperature | 0.7 |
| Top-p | 0.95 |
| Generations per prompt | 5 |
| Student model hyperparameter for training | |
| NVIDIA A6000 GPUs | 4 |
| Learning rate | $2 \times 10^{-5}$ |
| Epochs | 3 |
| Train batch size per device | 12 |
| Gradient accumulation steps | 2 |
| BF16 precision | False |
| PEFT Training | False |
| Train on correct gens only | True |

Table 5: Configurations and hyperparameters for SFT.

| Hyperparameter | |
| --- | --- |
| NVIDIA A6000 GPUs | 4 |
| Learning rate | $1 \times 10^{-6}$ |
| Epochs | 3 |
| Train batch size per device for 0.5B | 2 |
| Train batch size per device for $> 0.5$B | 1 |
| Generations per prompt | 4 |
| Max completion length | 2048 |
| $N_{\text{step}}$ for MATH | 32 |
| $N_{\text{step}}$ for MBPP+ and MiniF2F | 4 |
| BF16 precision | True |
| KL coefficient (for GRPO only) | 0.04 |
| RAPID Importance Weight Clipping $\eta$ | 2.0 |
| Colocate mode[1] | True |

Table 6: Configurations and hyperparameters for RL algorithms.

| Package | Version |
| --- | --- |
| python | 3.11.11 |
| trl | 0.24.0.dev0 |
| vllm | 0.8.1 |
| pytorch | 2.6.0 |

Table 7: Package versions.

---

[1]New in the latest TRL library for GPU alternation between inference and backpropagation.

```
compute_environment: LOCAL_MACHINE
debug: false
deepspeed_config:
  zero3_init_flag: false
  zero3_save_16bit_model: false
  zero_stage: 2
  gradient_clipping: auto
distributed_type: DEEPSPEED
machine_rank: 0
main_training_function: main
num_machines: 1
num_processes: 4
rdzv_backend: static
same_network: true
tpu_env: []
tpu_use_cluster: false
tpu_use_sudo: false
use_cpu: false
```

Figure 5: DeepSpeed configuration.

