# OpenReview forum: "RAPID: An Efficient Reinforcement Learning Algorithm for Small Language Models"
_ICLR.cc/2026/Conference — Submitted to ICLR 2026_

### Official Review · Reviewer_Zadd · 2025-10-20

**Soundness:** 2
**Presentation:** 2
**Contribution:** 2
**Rating:** 2
**Confidence:** 4

**Summary:**

This paper proposes an RL algorithm, RAPID, which during each iteration first generates a large amount of samples and conduct offline policy gradient with group-relative rewards. Importance weightings are adopted for gradient estimation. The algorithm can save runtime compared with existing baselines like GRPO and PG. The experiments are conducted on three benchmarks, comparing RAPID with various baselines. The authors also provide analysis on the role of training batch size.

**Strengths:**

- Clear illustration of the algorithm.
- Extensive experiments on various benchmarks.
- The runtime is reduced compared with existing baselines.

**Weaknesses:**

- The figure 1 is a bit misleading. As in real experiments (say table 1), RAPID cannot save as much as 89% runtime compared with GRPO. And the number 89% only appears once, which is a bit confusing for readers.
- The reviewer cannot see much novelty in the algorithm design. It's more like a variant of GRPG (a simplified GRPO) with offline and iteratively generated batches, since all implementations are common practice in GRPO [1] and off-policy policy gradient [2].
- RAPID cannot outperform the simplest RL algorithm, PG, on Math and MiniF2F (two of three benchmarks in table 1), and the runtime advantage is not prominent on MATH compared with PG. Table 2 reflects the same trend on Qwen and llama.
- Figure 2 is noisy and cannot deliver convincing information.
- Hyperparameter H requires to be carefully tuned for good performance. This drawback further decreases the training efficiency.

[1] Shao et al. DeepSeekMath: Pushing the Limits of Mathematical Reasoning in Open Language Models. https://arxiv.org/pdf/2402.03300

[2] https://lilianweng.github.io/posts/2018-04-08-policy-gradient/#off-policy-policy-gradient

**Questions:**

- Could you compare the runtime memory usage of these methods? RAPID would use a larger batch size for inference.

---

> ### Author Response · Authors · 2025-12-03
>
> Thank you very much for your feedback! We wanted to address your questions and concerns as follows:
>
> >_“The reviewer cannot see much novelty in the algorithm design. It's more like a variant of GRPG (a simplified GRPO) with offline and iteratively generated batches, since all implementations are common practice in GRPO [1] and off-policy policy gradient [2].”_
>
> GRPO and off-policy policy gradient algorithms can use offline generated batches, but their method is to do multiple gradient steps on the same samples. Our method samples ahead for multiple batches, and backpropagates on each one of them (once), while correcting for staleness. The multi-batch sampling is what amortizes sampling time, and thus overall training time of the algorithm.
>
> >_“RAPID cannot outperform the simplest RL algorithm, PG, on Math and MiniF2F (two of three benchmarks in table 1), and the runtime advantage is not prominent on MATH compared with PG. Table 2 reflects the same trend on Qwen and llama.”_
>
> The sacrifice to accuracy on Math and MiniF2F compared to PG is relatively mild (~1%), especially considering the size of the evaluation datasets. Our runtime advantage on MATH is 12% improvement compared to PG.

---

### Official Review · Reviewer_jQi7 · 2025-10-25

**Soundness:** 2
**Presentation:** 1
**Contribution:** 1
**Rating:** 0
**Confidence:** 5

**Summary:**

The paper proposes RAPID (Reweighted Advantage for Preemptive Inference of Data), a novel reinforcement learning (RL) algorithm for small language models (SLMs) to reduce training time during fine-tuning (for tasks like math, coding, and formal theorem-proving) while maintaining or improving accuracy; its core design includes alternating between large-batch inference (leveraging the memory efficiency of inference servers like vLLM to maximize efficiency) and small-batch off-policy policy gradient updates (optimizing gradient descent efficiency), integrating group advantage estimation into the policy gradient framework, and deriving an importance weighted estimator to correct off-policy learning bias—avoiding the performance limitations of KL-divergence regularization used in methods like PPO when taking multiple off-policy steps; the paper also conducts extensive experiments on three benchmarks (MBPP+ for coding, MATH for mathematics, MiniF2F for formal theorem-proving) using SLMs such as Qwen2.5, Llama 3.2 1B, and Gemma 3 1B, comparing with baselines like SFT, GRPO, Policy Gradient, and DAPO, and the results show RAPID reduces training time by 11%–34% across datasets (34% on MBPP+, 32% on MiniF2F, 11% on MATH) while achieving similar or better accuracy,

**Strengths:**

- The work focuses on an essential problem in reinforcement learning optimization, which is highly relevant for scaling RL algorithms to large models and complex environments.

- The manuscript is clearly presented, and the experimental section is complete enough.

**Weaknesses:**

1. The paper lacks novelty. The proposed components appear to be borrowed from existing popular approaches.

- The idea of inference in large batches has already been implemented in modern inference engines such as vLLM, which can automatically adjust batch sizes to the maximum available capacity.

- The use of group advantage estimation with normalization has been discussed in prior works such as GRPO and Kimi K1.5, and has already been widely adopted in subsequent research.
Therefore, the methodological contribution of this paper seems limited.

2. There are concerns regarding the correctness and credibility of the implementation and reported results. In Figure 1, the performance drop shown is excessively large, while the reported rollout time appears unrealistically low. If the only modification is increasing batch size, the time consumption should decrease partially but not be eliminated entirely. This raises doubts about whether the implementation and experimental measurements are accurate.

**Questions:**

Please see problem in section Weaknesses.

---

> ### Author Response · Authors · 2025-12-03
>
> Thank you very much for your feedback! We wanted to address your questions and concerns as follows:
>
> >_“The paper lacks novelty. The proposed components appear to be borrowed from existing popular approaches. The idea of inference in large batches has already been implemented in modern inference engines such as vLLM, which can automatically adjust batch sizes to the maximum available capacity.”_
>
> There is indeed the option to adjust and maximize batch size for inference engines such as vLLM, but the backpropagation step in GRPO has a much lower capacity, meaning one cannot naively maximize inference batch size for GRPO. This mismatch in optimal/maximum batch size for inference and backpropagation is exactly the problem our algorithm is trying to solve—one can still sample in a large batch, making it closer to the optimum, but we can backpropagate in smaller batches and correct for sample staleness. We show that this accelerates training while having little to no sacrifice to accuracy.
>
> >_“The use of group advantage estimation with normalization has been discussed in prior works such as GRPO and Kimi K1.5, and has already been widely adopted in subsequent research. Therefore, the methodological contribution of this paper seems limited.”_
>
> Our contribution is not group advantage estimation with normalization; instead, we build on it by proposing a novel way to accelerate the algorithm.
>
> >_“There are concerns regarding the correctness and credibility of the implementation and reported results. In Figure 1, the performance drop shown is excessively large, while the reported rollout time appears unrealistically low. If the only modification is increasing batch size, the time consumption should decrease partially but not be eliminated entirely. This raises doubts about whether the implementation and experimental measurements are accurate.”_
>
> Figure 1 does not illustrate the contribution of our paper—it only shows the inefficiency of a naïve GRPO implementation that uses the same inference and backpropagation batch size. Since the maximum possible backpropagation batch size is very low, this makes inference using the same batch size very inefficient. The drop in rollout time appears large only because the comparison is against a naïve implementation (though the new rollout time is still non-trivial). Our paper does not use this as our baseline, but instead, we compare with the state-of-the-art algorithms like DAPO.

---

### Official Review · Reviewer_gPE1 · 2025-10-31

**Soundness:** 2
**Presentation:** 2
**Contribution:** 1
**Rating:** 2
**Confidence:** 4

**Summary:**

This paper introduces RAPID (Reweighted Advantage for Preemptive Inference of Data), a novel reinforcement learning algorithm designed to enhance the efficiency of fine-tuning SLMs. The method alternates between large-batch inference and off-policy policy-gradient updates on smaller mini-batches, optimizing both inference and backpropagation phases. To address off-policy bias, RAPID employs an importance-weighted group advantage estimator and performs updates efficiently using stored experience data.

**Strengths:**

1. RAPID reduces RL training time, while maintaining competitive performance, making it especially valuable for resource-constrained deployment scenarios.

2. The alternating design elegantly decouples inference and training, and the importance-weighted group advantage estimator effectively mitigates off-policy bias.

3. The paper is well structured, with clear algorithmic exposition and empirical validation.

**Weaknesses:**

1. While runtime efficiency improves, accuracy gains over GRPO and PG are relatively modest, especially on the MATH and MiniF2F datasets (Table 1).

2. Evaluation is limited to three datasets (MBPP+, MATH, MiniF2F), which primarily cover mathematical and programming domains. Broader validation on diverse reasoning and language tasks (e.g., commonsense QA, dialogue, summarization) would strengthen generalizability.

3. The paper lacks ablation studies isolating the contributions of the importance weighting and group advantage estimation components. Such analyses would clarify their respective impacts on overall performance.

**Questions:**

1–3. See weaknesses above.

4. How sensitive is RAPID to the choice of importance weight clipping threshold $\eta = 2.0$ ? Was this parameter tuned per dataset, or is it robust across different tasks? An ablation study would be informative.

5. The algorithm introduces bias through importance weight clipping—has this bias been quantified empirically?

6. How does RAPID scale with larger models (e.g., 7B–13B)? Does the runtime advantage increase linearly with model size, or are there diminishing returns?

---

> ### Author Response · Authors · 2025-12-03
>
> Thank you very much for your feedback! We wanted to address your questions and concerns as follows:
>
> >_“While runtime efficiency improves, accuracy gains over GRPO and PG are relatively modest, especially on the MATH and MiniF2F datasets (Table 1).”_
>
> The overall goal of our algorithm is to improve runtime. We record accuracy to show that there is little to no sacrifice there.
>
> >_“How sensitive is RAPID to the choice of importance weight clipping threshold ? Was this parameter tuned per dataset, or is it robust across different tasks? An ablation study would be informative.”_
>
> The same threshold 2.0 is applied across the board, and we find that it is generally not necessary to tune it to datasets/models as the percentage of clipping is consistently low.

---

### Official Review · Reviewer_fjRE · 2025-10-31

**Soundness:** 1
**Presentation:** 3
**Contribution:** 1
**Rating:** 2
**Confidence:** 5

**Summary:**

This paper proposes RAPID, which focuses on reducing the running time of RL. Specifically, RAPID performs inference in large batches, and then performs off-policy policy gradient updates in mini-batches. Authors conducted experiments on MBPP+, MATH and MiniF2F with Qwen-2.5, 0.5B and 1.5B, Llama-3.2 1B, and Gemma-3 1B to illustrate the effectiveness of RAPID.

**Strengths:**

1. This paper is clearly written and easy to follow.

**Weaknesses:**

1. Although the authors claim in the title, abstract, and many other places throughout the paper that their primary focus is on small language models, I did not see any special design in their proposed algorithm, RAPID, that specifically targets them. Furthermore, the baselines they compare against, such as DAPO and GRPO, are also not algorithms specifically designed for small language models.
2. Authors claim that "We refer to the algorithm combining group advantage estimation and the policy gradient algorithm as group relative policy gradient (GRPG); this algorithm is a traditional on-policy RL algorithm", which is an erroneous statement. Equation (3) of the GPRO paper apparently considers importance sampling for off-policy updates.
3. Limited technical novelty. Authors claim the contribution of RAPID is doing inference at a large batch and then updating the policy with mini-batches to save computation. However, this is indeed the way that GRPO does! Please read carefully the open-source code of GRPO in verl.

**Questions:**

1. Could you please elaborate more on the value term in Equation (4)? From the theory of baseline, the baseline term can be anything as long as it is independent of the current policy. So I wonder the effectiveness of adding an additional IS term in value estimation.

---

> ### Author Response · Authors · 2025-12-03
>
> Thank you very much for your feedback! We wanted to address your questions and concerns as follows:
>
> >_“Limited technical novelty. Authors claim the contribution of RAPID is doing inference at a large batch and then updating the policy with mini-batches to save computation. However, this is indeed the way that GRPO does! Please read carefully the open-source code of GRPO in verl.”_
>
> The original GRPO algorithm and current open source implementations of it do have off-policy elements, but their method is to do multiple gradient steps on the same samples -- in particular, see lines 10-11 in Algorithm 1 in Shao et al. (2024), the paper that introduced GRPO. Our method samples ahead for multiple batches, and backpropagates on each one of them (once), while correcting for staleness. The multi-batch sampling is what amortizes sampling time, and thus overall training time of the algorithm.
>
>
> >_“​​Authors claim that "We refer to the algorithm combining group advantage estimation and the policy gradient algorithm as group relative policy gradient (GRPG); this algorithm is a traditional on-policy RL algorithm", which is an erroneous statement. Equation (3) of the GPRO paper apparently considers importance sampling for off-policy updates.”_
>
> GRPO is based on PPO, which is widely considered to be an on-policy RL algorithm. While it incorporates off-policy updates, these updates are typically used to perform multiple gradient steps on the same data, as described above. These updates can typically only be taken for a small number of steps. In contrast, our algorithm is designed to take gradient steps on data that is significantly more stale. We acknowledge that this is a gray area, but our terminology is consistent with the literature.
>
>
> >_“Could you please elaborate more on the value term in Equation (4)? From the theory of baseline, the baseline term can be anything as long as it is independent of the current policy. So I wonder the effectiveness of adding an additional IS term in value estimation.”_
>
> For actor-critic algorithms, the best baseline is an estimate of the value function (constant baselines can typically only be used with REINFORCE). We are using the former; thus, we need to use importance weighting to obtain an unbiased policy gradient estimate.

---

### Meta-Review · Area_Chair_XRUw · 2026-01-07

**Summary:**

Reviewers raised concerns about limited novelty (seems like a variant of GRPO/off-policy PG), modest accuracy gains on MATH/MiniF2F, narrow evaluation domains, and lack of ablation studies. The rebuttal clarified technical differences (multi-batch sampling vs. multiple gradient steps) but didn’t fully address novelty or evaluation breadth.

**Reviewer Concerns:**

gPE1’s concerns about accuracy and need for broader tasks remain outstanding; Zadd’s novelty concern is partially addressed but still valid. Both reviewers’ points about missing ablation studies and sensitivity analysis are not resolved.

**Reviewer Scores:**

gPE1: might move from 2→3 after rebuttal (clarifications help but core issues remain).

Zadd: likely stays at 2 (novelty still limited).

---

### Decision · Program_Chairs · 2026-01-26

Reject